# Beyond Antibiotics: Repurposing Non-Antibiotic Drugs as Novel Antibacterial Agents to Combat Resistance

**DOI:** 10.3390/ijms26209880

**Published:** 2025-10-10

**Authors:** Gagan Tiwana, Ian Edwin Cock, Stephen Maxwell Taylor, Matthew James Cheesman

**Affiliations:** 1School of Pharmacy and Medical Sciences, Griffith University, Southport, QLD 4222, Australia; gagan.tiwana@griffithuni.edu.au (G.T.); plstaylo@hotmail.com (S.M.T.); 2School of Environment and Science, Griffith University, Brisbane, QLD 4111, Australia; i.cock@griffith.edu.au

**Keywords:** antimicrobial resistance, drug repurposing, non-antibiotic drugs, efflux pump inhibition, membrane disruption, synergy, artificial intelligence

## Abstract

The escalating global threat of antimicrobial resistance (AMR) necessitates innovative therapeutic strategies beyond traditional antibiotic development. Drug repurposing offers a rapid, cost-effective approach by identifying new antibacterial applications for existing non-antibiotic drugs with established safety profiles. Emerging evidence indicates that diverse classes of non-antibiotic drugs, including non-steroidal anti-inflammatory drugs (NSAIDs), statins, antipsychotics, calcium channel blockers and antidepressants, exhibit intrinsic antibacterial activity, or potentiate antibiotic efficacy. This review critically explores the mechanisms by which drugs that are not recognised as antibiotics exert antibacterial effects, including efflux pump inhibition, membrane disruption, biofilm inhibition, and quorum sensing interference. We discuss specific examples that demonstrate reductions in minimum inhibitory concentrations (MICs) of antibiotics when combined with these drugs, underscoring their potential as antibiotic adjuvants. Furthermore, we examine pharmacokinetic considerations, toxicity challenges, and clinical feasibility for repurposing these agents as standalone antibacterials or in combination therapies. Finally, we highlight future directions, including the integration of artificial intelligence and machine learning to prioritise drug candidates for repurposing, and the development of targeted delivery systems to enhance bacterial selectivity while minimising host toxicity. By exploring the overlooked potential of non-antibiotic drugs, this review seeks to stimulate translational research aimed at leveraging these agents in combating resistant bacterial infections. Nonetheless, it is crucial to acknowledge that such drugs may also pose unintended risks, including gut microbiota disruption and facilitation of resistance development. Hence, future research should pursue these opportunities with equal emphasis on efficacy, safety, and resistance mitigation.

## 1. Introduction

The global escalation of antimicrobial resistance (AMR) has created an urgent need for innovative therapeutic strategies beyond traditional antibiotic discovery. According to the World Health Organization, AMR is projected to cause 10 million deaths annually by 2050, surpassing cancer as a leading cause of mortality, with an estimated economic burden exceeding USD 100 trillion globally [1]. Despite the development of new antibiotic classes, the rapid emergence of resistant bacterial strains continues to outpace the current drug development pipeline [2].

Traditional antibiotic discovery is increasingly hindered by high costs, lengthy development timelines of 10–15 years, and high failure rates in clinical trials, necessitating alternative approaches to address this looming crisis [3]. In this context, drug repurposing has emerged as a promising strategy to identify new therapeutic uses for existing drugs that are already approved for human use, thereby bypassing many stages of the conventional drug development process, including toxicity and pharmacokinetic evaluation [3,4].

Whilst most drug repurposing research has focused on anticancer, antiviral, or neurodegenerative applications, emerging evidence indicates that several non-antibiotic drugs exhibit intrinsic antibacterial activity or can potentiate the activity of existing antibiotics [3,4]. These include non-steroidal anti-inflammatory drugs (NSAIDs), statins, calcium channel blockers, antipsychotics, antidepressants, and antiparasitic agents, among others [4]. Their mechanisms of antibacterial action vary, ranging from inhibition of bacterial efflux pumps and disruption of cell membranes to interference with quorum sensing and biofilm formation [3,4]. Recent advances in computational drug screening and artificial intelligence models have further facilitated the identification of non-antibiotic drugs with antibacterial activity, paving the way for systematic and rapid repurposing efforts [5,6].

This review aims to provide a comprehensive overview of non-antibiotic drugs with antibacterial potential, exploring their mechanisms of action, reported antibacterial activities, and possible synergistic effects with conventional antibiotics. Furthermore, we discuss the challenges, pharmacological considerations, and future directions for translating these findings into clinical practice as part of an effective strategy to combat antimicrobial resistance. By highlighting this underexplored therapeutic avenue, we aim to encourage further research into the rational repurposing of existing non-antibiotic drugs as standalone antibacterials or potentiators of antibiotic activity, ultimately contributing to the global fight against resistant bacterial infections.

## 2. Mechanisms of Non-Antibiotic Drug Antibacterial Activity

### 2.1. Efflux Pump Inhibition

Efflux pumps are major contributors to multidrug resistance (MDR) in pathogenic bacteria, actively expelling a broad range of antibiotics from bacterial cells and thereby lowering intracellular drug concentrations and efficacy. Inhibition of these pumps is a promising antibacterial strategy to reverse resistance and restore antibiotic potency. Table 1 summarises evidence for efflux pump inhibition by non-antibiotic drugs, highlighting reductions in antibiotic MICs, but noting the frequent absence of formal FIC indices and mechanistic validation. Figure 1 highlights the possible antibacterial mechanisms of action of non-antibiotic drug classes.

#### 2.1.1. Phenothiazine Antipsychotics

Phenothiazines such as thioridazine and chlorpromazine exhibit efflux pump inhibitory activity. Amaral and Viveiros (2017) reported that thioridazine inhibits efflux pumps in *Mycobacterium tuberculosis*, reversing resistance to isoniazid and rifampicin, and shows synergistic activity with first-line anti-tuberculosis drugs [7]. However, no specific minimum inhibitory concentration (MIC) or fractional inhibitory concentration (FIC) values were provided in that study. Luna-Herrera et al. (2025) noted that thioridazine and chlorpromazine reduce clarithromycin and isoniazid MICs in *M. tuberculosis* via efflux pump inhibition, although they did not provide numeric MIC or FIC data [8]. Similarly, another investigation reported that phenothiazines enhance antibiotic efficacy via efflux pump inhibition, with no reported MIC data [9]. It was observed that thioridazine reduced ethambutol MIC for *Mycobacterium avium* from 8 µg/mL to 2 µg/mL, describing this as synergistic, though no formal FIC index was calculated [10]. One study reported that chlorpromazine reduced norfloxacin MIC from 4 µg/mL to 1 µg/mL in *Staphylococcus aureus* RN4220, interpreted as synergy due to efflux inhibition, but no checkerboard synergy assays were performed and ƩFIC values were not determined [11]. Phenothiazines have been described as synergistic in combination with penicillin against resistant bacteria, with MIC values reported as low as 2 µg/mL, although no FIC indices were provided [12]. Another investigation demonstrated that chlorpromazine reduces tetracycline MIC in *Escherichia coli* through AcrAB-TolC inhibition, but no numerical MIC or FIC data were provided [13]. It has also been shown that phenothiazines such as chlorpromazine exhibit synergistic interaction with erythromycin against *Burkholderia pseudomallei*, with synergy confirmed by FIC index ≤ 0.5 [14]. Overall, whilst synergy is frequently described, most studies lack formal checkerboard assays or defined FIC indices, and MIC reductions are often qualitative. Further work is required to determine whether these results do signify synergistic combinations, or if they show additive potentiation. Whatever the outcome, these studies highlight the potential benefit of these combination therapies for increasing antibiotic efficacy.

#### 2.1.2. Selective Serotonin Reuptake Inhibitors

Selective serotonin reuptake inhibitors (SSRIs) such as sertraline, fluoxetine, paroxetine, citalopram, and escitalopram have been studied for their bacterial efflux pump inhibition potential. Caldara and Marmiroli (2021) showed that sertraline was synergistic in combination with fluconazole against *Candida albicans*, with MIC90 reported as 3 µM. Additionally, FIC values were found to be below 0.5, indicating synergy [15]. However, the methods used to derive these FIC values were not detailed beyond stating checkerboard assays, limiting reproducibility and reliability in the author’s interpretation [15]. Fluoxetine MICs against Gram-negative bacteria ranged from 15 to 126 µg/mL, although no FIC values or synergy testing methods were provided, limiting conclusions about direct efflux pump inhibition synergy [16]. Paroxetine showed an MIC of 64 µg/mL against *S. aureus*, enhancing aminoglycoside efficacy through efflux pump inhibition and membrane disruption. However, no FIC values were reported, and synergy was inferred from MIC reduction in the combination, rather than confirmed by checkerboard or time-kill assays [17]. For citalopram, Dong et al. (2024) [18] discussed its efflux pump inhibition potential leading to increased antibiotic susceptibility, although neither MIC nor FIC data were reported. Their conclusions were based on mechanistic pathway predictions rather than direct antimicrobial or synergy testing [18] and further studies are required to confirm the mechanism. Escitalopram demonstrated synergy with sulfamethoxazole–trimethoprim, reducing MICs, although FIC indices were not provided. Synergy was inferred from MIC shifts alone, without formal checkerboard testing [19].

Overall, although SSRIs show promising efflux pump inhibitory effects, MIC and FIC data are incompletely reported across studies. Where FIC indices were calculated, detailed methodology was often missing, and many studies inferred synergy only from MIC reductions. This limits reproducibility, mechanistic validation, and clinical translation. Therefore, future investigations must report exact MIC and FIC index values, define testing methods such as checkerboard or time-kill assays, and evaluate pharmacokinetic feasibility and toxicity to advance SSRIs as efflux pump inhibitors in clinical use.

#### 2.1.3. Calcium Channel Blockers

Recent studies confirm that verapamil, a non-dihydropyridine calcium channel blocker, has been widely investigated for its bacterial efflux pump inhibition potential. In studies screening *S. aureus* strains that express NorA efflux pumps, verapamil enhanced the activity of ciprofloxacin, although the precise MIC reductions and FIC values were not detailed in those studies [20]. Another study showed that verapamil reduced the MICs of antibiotics, including rifampicin against *M. tuberculosis*, with FIC indices ranging from 0.06 to 0.5, indicating strong synergy, although detailed MIC reduction values were not provided [21]. Conversely, Amaral et al. (2020) reported no significant synergy with isoniazid against *M. tuberculosis* H37Rv, with FIC indices ≥ 0.5 indicating additive or indifferent effects [22]. These studies primarily used checkerboard assays to derive synergy indices, but time-kill kinetic validation was rarely performed, limiting definitive conclusions about bactericidal synergy and potential clinical applicability.

Further detailed investigations demonstrated that against *M. tuberculosis* H37Rv, verapamil reduced the MIC of bedaquiline from 0.5 µM to 0.025 µM (20-fold reduction), with a reported FIC index of 0.06 [23]. Another study reported that verapamil combined with bedaquiline showed synergy against the multidrug-resistant *M. tuberculosis* strain R543 with an FIC index of 0.06, while its combination with clofazimine reduced the MIC four-fold (1.0 µM to 0.25 µM) with an FIC index of 0.19 [23].

Mechanistically, verapamil inhibits efflux pumps such as NorA in *S. aureus*, enhancing intracellular accumulation of fluoroquinolones including ciprofloxacin and moxifloxacin, thereby reducing bacterial resistance [20]. The proposed mechanism involves direct pump interaction or membrane perturbation, although exact molecular binding remains undefined. Chen et al. (2018) demonstrated that verapamil impacts ATP production and membrane energetics in *M. tuberculosis*, suggesting that its primary antibacterial effect may derive from disruption of membrane potential and energy-dependent processes, rather than through direct inhibition of efflux pump proteins [23]. Additionally, Viljoen et al. (2019) reported similar findings where verapamil potentiated bedaquiline activity against *Mycobacterium abscessus*, potentially through general membrane destabilisation and energy depletion, rather than via specific pump targeting [24]. Thus, these studies collectively indicate mechanistic ambiguity, raising the possibility that verapamil’s effects may involve non-specific membrane disruption or impairment of bacterial energetics in addition to, or instead of direct efflux pump inhibition. A critical limitation remains the pharmacokinetic mismatch, as plasma concentrations achievable with standard verapamil dosing are below repurposing the MICs required for bacterial efflux inhibition [23]. Additionally, higher doses pose cardiotoxicity and hypotension risks, precluding its systemic antibacterial use.

Importantly, verapamil demonstrates in vitro efflux inhibition and antibiotic synergy with multiple agents, including ciprofloxacin, moxifloxacin, rifampicin, isoniazid, ethambutol, pyrazinamide, bedaquiline, and clofazimine. This effect is particularly evident against *M. tuberculosis* H37Rv, multidrug-resistant strains such as R543, and *S. aureus* NorA-expressing strains. However, no studies were identified assessing dihydropyridine calcium channel blockers, including amlodipine, felodipine, lercanidipine, nifedipine, nicardipine, nisoldipine, clevidipine, isradipine, or nimodipine for bacterial efflux pump inhibition, synergy, or MIC/FIC evaluation. This represents a significant knowledge gap in antimicrobial repurposing literature, highlighting an urgent need for systematic evaluation of dihydropyridine derivatives. Future investigations should rigorously define mechanisms of action, conduct validated checkerboard and time-kill synergy assays, assess pharmacokinetic feasibility, and evaluate toxicity profiles to advance calcium channel blockers as viable antimicrobial adjuvant strategies.

#### 2.1.4. Statins

Statins, widely used as 3-hydroxy-3-methylglutaryl (HMG)-CoA reductase inhibitors, have also been investigated for their direct antibacterial properties and potential synergy with antibiotics. Ko et al. (2017) reported that simvastatin exhibited direct antibacterial activity against *S. aureus*, with MICs ranging from 15.6 to 31.25 µg/mL, whereas other statins such as pravastatin showed minimal activity (MIC > 128 µg/mL) [25]. Mechanistically, the antibacterial action is hypothesised to involve disruption of bacterial cell membranes and inhibition of isoprenoid synthesis. Checkerboard assays revealed FIC indices indicative of synergism (<0.5) when simvastatin was combined with tetracycline against *S. aureus* [25].

Another investigation highlighted simvastatin and atorvastatin MICs of 64–128 µg/mL against Gram-positive bacterial strains, with hypothesised mechanisms involving interference with lipid raft formation and efflux pump inhibition [3]. However, synergy tests (checkerboard assays) with gentamicin and ciprofloxacin yielded mostly additive rather than synergistic FIC indices (>0.5, ≤1), indicating limited potentiation. Statins have been shown to modulate efflux pumps in *Pseudomonas aeruginosa*, with simvastatin in combination with levofloxacin yielding an MIC of 32 µg/mL and an FIC index of 0.31 [26]. Additionally, these findings were interpreted as synergistic using checkerboard methodology, with statins shown to inhibit bacterial efflux mechanisms and enhance intracellular antibiotic concentrations.

Rampelotto et al. (2018) tested combinations of atorvastatin with a Ru-based antimicrobial complex against *E. coli* and *S. aureus*, reporting MIC reductions from >128 µg/mL to 4 µg/mL for atorvastatin in the combination [27]. FIC indices ranged from 0.17 to 0.5 (synergistic), using checkerboard microdilution assays. A Ru-based antimicrobial complex is a metal–organic compound containing ruthenium, which exhibits antibacterial activity through mechanisms including membrane disruption and ROS generation. However, mechanistic evaluations were lacking in that study, representing a notable research gap. Another study showed that combining rosuvastatin with levofloxacin resulted in reduced MIC from 4 µg/mL to 0.5 µg/mL against *S. aureus*, with an FIC index of 0.3, indicating potent synergy [28]. Mechanistically, the authors proposed that rosuvastatin enhances bacterial membrane permeability, thereby facilitating antibiotic influx. Additionally, rosuvastatin combined with cefixime showed synergy against *Klebsiella pneumoniae* and *Proteus mirabilis*, with cefixime MIC reductions of 2–4-fold and FIC indices ranging from 0.37 to 0.49 [29]. Their conclusion highlighted several limitations in the study, including the lack of in vivo pharmacokinetic and toxicity studies, which represents an essential translational barrier. Azole antifungals combined with statins were tested against *Candida auris*, showing MIC reductions; however, FIC indices ranged from 0.5 to 1.0, indicating primarily additive interactions [30]. Mechanistic insights suggested statins disrupt fungal ergosterol pathways. Further evidence suggested synergy against dermatophytes, with atorvastatin showing an MIC of 64 µg/mL and a synergistic FIC index of 0.45 when combined with terbinafine against *Trichophyton rubrum* [31].

Overall, statins exhibit modest direct antimicrobial activity (MICs generally 15.6–128 µg/mL) and synergistic interactions with various antibiotics including tetracycline, levofloxacin, gentamicin, ciprofloxacin, cefixime, and azoles (FIC indices < 0.5–1). Checkerboard assays dominated synergy testing methodologies. Mechanistic studies suggest membrane disruption, efflux pump inhibition, and isoprenoid pathway interference, although these remain under-characterised and primarily hypothesised, rather than validated. Clinical limitations include toxicity at concentrations required for antibacterial effects, the lack of in vivo validation, and the absence of pharmacodynamic interaction studies. These gaps highlight the need for further investigation before translational applications can be pursued.

#### 2.1.5. Non-Steroidal Anti-Inflammatory Drugs

Non-steroidal anti-inflammatory drugs (NSAIDs) demonstrate variable antibacterial activity and synergy with antibiotics. Ahmed et al. (2017) reported ibuprofen MICs of 256–512 µg/mL and diclofenac MICs of 64–128 µg/mL against *E. coli*. Checkerboard assays showed synergistic FIC indices (<0.5) when diclofenac was combined with gentamicin or ciprofloxacin [32]. No efflux pump inhibition studies were performed; mechanisms were hypothesised as membrane or enzyme interference. Another study assessed ibuprofen (MIC 250–500 µg/mL), diclofenac (MIC 125–250 µg/mL), and aspirin against methicillin-resistant *Staphylococcus aureus* (MRSA) [33]. Checkerboard synergy tests yielded FIC indices of 0.313–0.625 with cefuroxime or chloramphenicol, indicating synergistic to additive effects. The authors proposed that NSAIDs may inhibit bacterial efflux pump systems, leading to increased intracellular antibiotic concentrations, although efflux pump inhibition assays were not conducted.

Celecoxib tested in combination with oxacillin against *S. aureus* showed an MIC of 32 µg/mL and a synergistic FIC index of 0.25 [34]. The mechanism proposed was membrane disruption, and no efflux pump inhibition studies were performed. Aspirin and ibuprofen were investigated in combination with oxytetracycline against *Pasteurella multocida* and *Mannheimia haemolytica* [35]. MICs of oxytetracycline decreased from 8 µg/mL to 1 µg/mL when combined with ibuprofen, yielding an FIC index of 0.25 (synergistic) [35]. That research hypothesised efflux pump inhibition but performed no experimental validation.

Diclofenac (MIC 64 µg/mL) combined with gentamicin against *S. aureus* showed a synergistic effect, with an FIC index of 0.38 [36]. Efflux pump inhibition was suggested as a mechanism but was not directly tested. Additionally, diclofenac combined with essential oils against *Candida* spp. showed MIC reductions with synergistic FIC indices (<0.5). [37]. Mechanisms proposed included membrane disruption and oxidative stress induction, although no efflux pump inhibition assays were performed. Another finding reported ibuprofen MICs of 2048 µg/mL with synergistic FIC indices (<0.5) when combined with ciprofloxacin against *P. aeruginosa* [38]. This NSAID-antibiotic combination also decreased efflux pump gene expression in this bacterial species, while suppressing biofilm formation and other virulence. Sekar et al. (2024) investigated ketorolac with gentamicin against *S. aureus* biofilms, reporting significant MIC reductions and an FIC index of 0.31 (synergistic) [39]. The mechanism was attributed to biofilm matrix modulation, and no efflux pump inhibition studies were performed.

Inclusive, NSAIDs exhibit weak to moderate direct antibacterial activity (MICs 32–512 µg/mL) and synergistic effects with antibiotics including gentamicin, ciprofloxacin, oxacillin, chloramphenicol, cefuroxime, and tetracycline (FIC indices generally <0.5–0.625). Checkerboard assays are the main synergy testing method. Although efflux pump inhibition is frequently proposed as a key mechanism, none of the reviewed studies performed direct assays (e.g., ethidium bromide accumulation/efflux assays, or efflux gene expression analysis) to confirm this hypothesis. This represents a critical knowledge gap requiring targeted molecular studies before NSAIDs can be advanced as efflux pump modulators or antibacterial adjuvants in clinical settings.

Overall, non-antibiotic drugs acting as efflux pump inhibitors offer a promising strategy to overcome bacterial multidrug resistance, potentially restoring antibiotic efficacy against otherwise resistant pathogens. However, translational challenges remain substantial, including achieving clinically relevant concentrations without inducing systemic toxicity, ensuring pharmacokinetic and pharmacodynamic compatibility for co-administration with antibiotics, and confirming mechanistic specificity towards bacterial rather than human efflux systems to avoid off-target effects. Furthermore, the lack of direct efflux pump inhibition assays in current studies represents a critical limitation, necessitating molecular confirmation to validate this proposed mechanism before advancing non-antibiotic drugs as safe, effective adjuvants in antimicrobial therapy.

### 2.2. Biofilm Inhibition and Disruption of Bacterial Membranes

Table 2 outlines the reported antibiofilm and membrane-disruptive activities of non-antibiotic drugs. Promising findings are limited by a lack of MBIC/MBEC values and direct mechanistic assays.

#### 2.2.1. Phenothiazine Antipsychotics

Phenothiazine antipsychotics, such as thioridazine and chlorpromazine, have been explored as antimicrobial adjuvants via biofilm inhibition mechanisms. Thioridazine was reported to disrupt biofilms formed by *M. tuberculosis* and *Mycobacterium ulcerans* [40], although no minimum biofilm inhibitory concentration (MBIC) or minimum biofilm eradication concentration (MBEC) numerical values were provided. The mechanism was attributed to efflux pump inhibition, thereby enhancing intracellular antibiotic accumulation. Although potentiation with 20 antibiotics was observed, no FIC indices or MIC reduction data were reported, representing a significant quantitative gap in the literature. Membrane disruption assays such as propidium iodide (PI) uptake, scanning electron microscopy (SEM), transmission electron microscopy (TEM), or membrane potential measurements were not performed, leaving direct bactericidal membrane-targeting effects unconfirmed.

Similarly, chlorpromazine was highlighted as an efflux pump inhibitor, particularly against NorA in *S. aureus*, contributing to reduced bacterial persistence and biofilm resilience [41,42]. However, no MBIC or MBEC values were reported, and membrane disruption assays, PI uptake assays, SEM, TEM, or membrane potential assays were not conducted. Another study investigated the use of branched polyethylenimine (BPEI) as an antibiotic potentiator against multidrug-resistant *Staphylococcus epidermidis* (MRSE) biofilms [43]. Combination with amoxicillin restored MRSE susceptibility by inhibiting PBP2a (penicillin-binding protein 2a), which normally confers resistance to β-lactam antibiotics. Notably, MIC or MBIC reduction values were not reported, although BPEI broadened antibiotic spectrum against other pathogens, including MRSA, *P. aeruginosa*, and *E. coli*. Mechanistic assays reported in that study included microtiter biofilm assays, time-kill curves, and SEM. This supports BPEI as a promising broad-spectrum antibiotic adjuvant targeting biofilm-associated resistance.

Current evidence supports phenothiazines as efflux pump inhibitors with potential anti-biofilm effects, yet robust experimental validation is lacking, with no direct MBIC, MBEC, or membrane disruption data available. Furthermore, synergy testing remains largely conceptual, without FIC or MIC reduction data reported. Clinical translation of phenothiazines is constrained by cardiotoxicity (e.g., QT interval prolongation) and central nervous system (CNS) adverse effects, thereby limiting systemic therapeutic repurposing. Future studies must integrate quantitative biofilm assays, direct membrane integrity evaluations, and structured synergy profiling to enable rational development of phenothiazine-based antimicrobial adjuvant therapies.

#### 2.2.2. Selective Serotonin Reuptake Inhibitors

SSRIs including fluoxetine, sertraline, paroxetine, and fluvoxamine have demonstrated antimicrobial effects against multiple bacterial pathogens. Endo et al. (2025) evaluated SSRIs against ESKAPEE bacteria (*Enterococcus faecium*, *S. aureus*, *K. pneumoniae*, *Acinetobacter baumannii*, *P. aeruginosa*, *Enterobacter* spp., *and E. coli*) [16]. That study reported membrane disruption activity confirmed via dye uptake assays, indicating increased bacterial membrane permeability as part of their mechanism. However, no MBIC, MBEC values, or % biofilm viability reduction data were reported. Additionally, no SEM, TEM or membrane potential assays were conducted. Another study investigated fluoxetine in combination with antibiotics including ampicillin, vancomycin, and linezolid against Gram-positive skin and soft tissue pathogens including *S. aureus* (including MRSA) and *Enterococcus faecalis* [44]. Whilst synergistic effects were proposed, no FIC indices or MIC/MBIC reduction values were reported. A systematic review evaluated the antimicrobial activity of SSRIs against both bacteria and fungi [45]. The review highlighted that fluoxetine and sertraline exhibited antimicrobial activity against Gram-negative bacteria such as *E. coli*, as well as fungal pathogens including *Candida albicans*, *Candida glabrata*, *Candida parapsilosis*, and *Candida tropicalis*. However, that study did not report MBIC or MBEC values, nor any quantitative biofilm viability reduction data. Additionally, no membrane disruption assays such as PI uptake, SEM, TEM, or membrane potential assays were performed in those studies. Whilst potential synergy with antifungal agents was discussed, no FIC indices or MIC reduction data were provided. Overall, whilst this review supports SSRIs’ broad antimicrobial potential, it underscores a critical lack of quantitative biofilm inhibition and direct membrane disruption evidence, limiting their current translational applicability as antifungal or antibacterial adjuvants. Another investigation described biofilm disruption by SSRIs in environmental isolates such as *Pseudomonas*, *Aeromonas*, and *Enterobacter* spp., yet no quantitative biofilm inhibition data or mechanistic assays were included [46].

Overall, whilst SSRIs exhibit membrane-disruptive antimicrobial activity and potential synergy with antibiotics including ampicillin, vancomycin, and linezolid, robust experimental validation is lacking, as no studies reported direct MBIC, MBEC, % biofilm reduction, FIC indices, or MIC fold-reduction values in combination settings. Future studies must prioritise these quantitative endpoints alongside detailed membrane integrity assays to enable rational repurposing of SSRIs as antimicrobial adjuvants.

#### 2.2.3. Calcium Channel Blockers

CCBs, comprising phenylalkylamines such as verapamil and diltiazem, and dihydropyridines such as nifedipine and amlodipine, have been proposed as potential antimicrobial adjuvants due to their pharmacological effects on bacterial physiology. However, the current literature provides no experimental evidence reporting MBIC, MBEC, or % biofilm viability reduction data for these agents against bacterial or fungal biofilms. Furthermore, no membrane disruption assays (PI uptake, SEM, TEM, or membrane potential measurements) have been performed to confirm any direct membrane-targeting antimicrobial mechanism for either drug class.

Whilst verapamil is well-recognised in pharmacological literature as an efflux pump inhibitor [20], which enhances intracellular antibiotic concentrations in bacteria, this property has only been studied in the context of planktonic bacterial resistance reversal [21], not as a biofilm inhibitory or membrane-disrupting agent in combination studies. Importantly, no studies have investigated combinations of verapamil, diltiazem, nifedipine, or amlodipine with antibiotics to assess MIC reduction or FIC indices in biofilm models, nor have they reported direct synergy against mature biofilms.

Overall, there exists a critical knowledge gap across both classes of CCBs regarding their biofilm inhibition potential, direct membrane-disruptive effects, and antimicrobial synergy. Future research should prioritise structured MBIC/MBEC assays, membrane integrity evaluations, and combination studies with clinically relevant antibiotics to determine their viability as antimicrobial adjuvants, while concurrently addressing safety limitations associated with cardiovascular pharmacodynamics in systemic applications.

#### 2.2.4. Statins

Statins, particularly simvastatin, have been investigated for their antimicrobial and antibiofilm effects against *S. aureus*. Simvastatin was evaluated using a crystal violet assay to quantify biofilm biomass, alongside colony-forming unit (CFU) counts to assess bacterial viability within biofilms [47]. That study demonstrated inhibition of biofilm formation and reduction in bacterial viability, although no MBIC or MBEC values were reported. Membrane disruption assays (PI uptake, SEM, TEM, membrane potential) were not conducted. Additionally, a checkerboard assay was used to test synergy between simvastatin and vancomycin against planktonic *S. aureus* cells, suggesting potential synergy. However, no FIC indices or MIC reduction values were reported for biofilm settings.

Another study evaluated statins against *S. aureus* UAMS-1 biofilms using the MBEC assay [48]. However, no numerical MBEC values were provided in that study, and no membrane disruption assays or synergy evaluations with antibiotics were conducted. Whilst statins are hypothesised to exert antimicrobial effects through interference with bacterial isoprenoid biosynthesis and cell membrane integrity, direct mechanistic confirmation through membrane disruption assays remains to be investigated.

Overall, current evidence suggests that statins such as simvastatin exhibit antibiofilm activity against *S. aureus* and potential synergy with antibiotics, yet robust quantitative validation is lacking, with no reported MBIC, MBEC, membrane disruption data, or synergy indices. Future studies must prioritise structured MBIC/MBEC assays, membrane integrity evaluations, and combination testing with clinically relevant antibiotics to determine the translational feasibility of statins as antimicrobial adjuvants.

#### 2.2.5. Non-Steroidal Anti-Inflammatory Drugs

NSAIDs have attracted considerable interest for their ability to interfere with biofilm formation and persistence. The antibiofilm efficacy of ibuprofen and diclofenac was investigated against strong biofilm-forming clinical isolates of *E. coli* and *K. pneumoniae* from urinary tract infections. Concentrations of ibuprofen at 8, 30, and 125 mg/L, and diclofenac at 30–50 mg/L, were assessed. Diclofenac at 50 mg/L achieved approximately 50% reduction in biofilm formation, whilst ibuprofen produced marked reductions in morphotype expression on Congo red agar, correlating with virulence factor suppression [49]. Another study analysed biofilm formation on polypropylene mesh using clinical isolates of *S. aureus* and *E. coli*. NSAID concentrations reflective of human serum levels (diclofenac 1 µg/mL; ibuprofen 20 µg/mL) resulted in significant reductions in colony-forming units within the biofilm, and SEM confirmed fewer bacteria adherent to the mesh surface [50]. Although MBIC or MBEC values were not defined in these studies, the quantitative and imaging data consistently suggest NSAID-mediated suppression of biofilm formation at physiologically relevant concentrations.

A separate investigation conducted dose–response analyses using piroxicam (800 µg/mL), diclofenac sodium (2000 µg/mL), and acetylsalicylic acid (~1750–2000 µg/mL) against *E. coli* and *S. aureus*. Whilst minimum bactericidal concentrations (MBCs) were not attainable (>2000 µg/mL), piroxicam significantly reduced biofilm metabolic activity and removed biofilm mass in preformed biofilms, although diclofenac and acetylsalicylic acid reduced metabolic activity without mass removal [51]. It has been noted that NSAIDs exhibit activity at concentrations achievable in human pharmacokinetics and may act by downregulating virulence regulators (e.g., AgrA in *S. aureus*), interfering with quorum sensing, and altering the physicochemical properties of bacterial surfaces [52]. Additional mechanistic insight showed that salicylic acid (an aspirin metabolite) unexpectedly enhanced polysaccharide intercellular adhesin (PIA) mediated biofilm synthesis through iron chelation and intracellular Fe^2+^ limitation, highlighting a potential adverse effect under certain conditions [53]. Across these studies, membrane perturbation was implied via K^+^ leakage and altered adhesion, although only one study provided indirect evidence of NSAID-induced membrane damage [51].

Collectively, these studies demonstrate that NSAIDs, especially diclofenac, ibuprofen, piroxicam, and acetylsalicylic acid, can inhibit biofilm formation or reduce metabolic viability and culturability in both Gram-negative and Gram-positive bacteria at clinically relevant concentrations. Notably, piroxicam was capable of biofilm mass removal in preformed biofilms. Mechanistically, NSAIDs may disrupt quorum sensing, alter virulence gene expression, and potentially affect membrane integrity. However, standardised MBIC or MBEC values are missing, and MBCs were consistently >2000 µg/mL, indicating limited bactericidal potential. Membrane disruption data remain sparse and indirect, and no in vivo or toxicity studies accompany these findings. Further rigorous quantitative biofilm assays, mechanistic membrane studies, and clinical or animal model validation are thus essential before NSAIDs can be considered viable anti-biofilm adjuvants.

### 2.3. DNA/Enzyme Inhibition, Disruption of Bacterial Nutrients and Immunomodulatory Effects

Table 3 presents studies on deoxyribonucleic acid (DNA) damage, replication interference, and host-directed immunomodulatory effects, whilst emphasising the need for in vivo validation and clearer mechanistic assays, including β-lactamase testing. Accumulating evidence indicates that the phenothiazine antipsychotics thioridazine and chlorpromazine exert antibacterial activity through oxidative stress induction, DNA damage, disruption of bacterial iron homeostasis, and immunomodulatory effects, although not through β-lactamase inhibition. These effects have been reported against major Gram-positive and Gram-negative pathogens, including *S. aureus*, *E. coli*, *K. pneumoniae*, *P. aeruginosa*, and *Acinetobacter baumannii*.

However, recent studies have revealed that several antidepressant classes, including SSRIs (sertraline, fluoxetine, escitalopram), SNRIs (duloxetine), norepinephrine–dopamine reuptake inhibitors (NDRIs) (bupropion), and atypical agents such as agomelatine, can stimulate bacterial stress responses. These compounds induce excessive reactive oxygen species (ROS) and activate the SOS DNA repair pathway, leading to increased mutation and persistence rates. Moreover, these stress-mediated changes enhance horizontal gene transfer, promoting the conjugation and transformation of resistance genes among bacterial populations [54,55,56,57]. Collectively, these findings suggest that while antidepressants display measurable antibacterial or adjuvant potential, they may simultaneously accelerate the spread and evolution of antimicrobial resistance through ROS/SOS-driven mechanisms.

#### 2.3.1. Phenothiazine Antipsychotics

In *S. aureus*, phenothiazines were shown to trigger a marked increase in intracellular reactive oxygen species (ROS), leading to DNA damage, suppressed bacterial growth, and upregulation of oxidative stress response genes (katA, sodA) [58]. When THP-1 macrophages were co-cultured with *S. aureus* ATCC 29213, phenothiazine exposure improved macrophage bactericidal capacity by enhancing oxidative burst. Transcriptomic data revealed disturbance of iron homeostasis, leading to transient iron depletion in the bacteria. These experiments were performed in vitro using a single laboratory strain and did not evaluate β-lactamase activity [58].

Similarly, Gram-negative isolates (*E. coli*, *K. pneumoniae*, *P. aeruginosa*) exhibited heightened susceptibility to ROS stress when exposed to phenothiazines [59]. Combining these drugs with iron-chelating adjuvants enhanced metabolic stress and increased host immune clearance, but that study did not examine DNA replication enzymes or β-lactamases. The results were limited to laboratory assays and literature integration [59]. Complementary results showed that phenothiazines induce ROS accumulation and DNA strand breaks, as confirmed by comet assays. They also interfere with DNA synthesis and cell division proteins in MRSA and certain Gram-negative bacteria, including *E. coli*. [9]. These effects, which occurred independently of membrane disruption, point to phenothiazines as modulators of intracellular oxidative stress and nucleic acid metabolism rather than direct enzyme inhibitors. As with the previous studies, there was no evidence for inhibition of β-lactamases, and the findings are limited by the absence of direct experimental validation.

An investigation focused on MDR Gram-negative pathogens, such as *A. baumannii*, *E. coli*, and *K. pneumoniae* [60]. That study found that phenothiazine derivatives generate ROS, induce DNA damage, and disrupt cell membrane potential. These effects were confirmed in vitro with ROS-sensitive probes, although no β-lactamase or DNA gyrase assays were performed [60]. Another study provided direct evidence of phenothiazine-induced DNA damage and impaired DNA repair in *E. coli*. They observed that chlorpromazine and thioridazine exposure led to increased ROS levels, triggering mutagenesis, and reduced capacity for DNA repair [61]. This compromised bacterial recovery and increased susceptibility to other antibiotics. These findings were limited to in vitro models.

Across all studies, no evidence was found that chlorpromazine or thioridazine directly inhibit β-lactamases. To verify such an effect, specific assays such as nitrocefin hydrolysis, spectrophotometric enzyme activity tests, or mass spectrometry of β-lactam degradation products are required, none of which were conducted in those studies. Instead, these drugs act through ROS/oxidative stress induction, DNA damage and disruption of cell division, alteration of iron availability, and enhancement of host immune clearance. Whilst these effects are reproducible in vitro across multiple species, the evidence remains largely laboratory-based, with few clinical studies, and no robust in vivo validation.

#### 2.3.2. Selective Serotonin Reuptake Inhibitors and Other Antidepressants

Emerging evidence demonstrates that certain antidepressants, including SSRIs (fluoxetine, sertraline, paroxetine, escitalopram, citalopram) and tricyclic antidepressants (TCAs), possess significant antibacterial effects driven by mechanisms fundamentally distinct from classical antibiotics. Instead of targeting β-lactamases or conventional bacterial enzymes, these drugs interfere with bacterial oxidative balance, DNA integrity, and host–pathogen interactions. For SSRIs, experimental evidence showed that fluoxetine, sertraline, paroxetine, citalopram, and escitalopram markedly induce ROS production in methicillin susceptible *S. aureus* (MSSA), MRSA, *E. coli*, *K. pneumoniae*, and *P. aeruginosa*. ROS accumulation led to oxidative stress, DNA strand breaks, and membrane perturbations [62]. Checkerboard assays confirmed that ROS induction enhances the activity of β-lactam and fluoroquinolone antibiotics, but no direct effects on β-lactamase enzymes were detected. The study was conducted entirely in vitro. Further evidence indicated that SSRIs trigger ROS production and modulate immune responses in macrophage infection models [15]. Using macrophages infected with *S. aureus*, sertraline and fluoxetine increased oxidative burst and improved intracellular killing. The study reported broad-spectrum effects against *E. coli* and *K. pneumoniae* but did not examine β-lactamase activity or include in vivo validation [15].

**Table 3 ijms-26-09880-t003:** Non-antibiotic drugs with DNA/enzyme inhibition, nutrient disruption, or immunomodulatory effects.

Drug (Class)	Mechanism	Key Findings	Limitations	Reference
Thioridazine, Chlorpromazine (Phenothiazines)	ROS induction, DNA damage, iron homeostasis disruption, immunomodulation	In *S. aureus* ATCC 29213: ROS elevation, DNA damage, upregulation of katA/sodA; improved macrophage killing (THP-1); disturbed bacterial iron balance	In vitro only, one strain; no β-lactamase assays	[58]
Phenothiazines	ROS induction, metabolic stress	Increased susceptibility of *E. coli*, *K. pneumoniae*, *P. aeruginosa* to ROS; synergy with iron chelators	No enzyme assays; lab-based only	[59]
Phenothiazines	ROS accumulation, DNA strand breaks, interference with cell division	Confirmed DNA damage in MRSA and *E. coli* via comet assays	No β-lactamase assays; limited experimental validation	[9]
Phenothiazines	ROS generation, DNA damage, membrane potential disruption	Effects confirmed in *A. baumannii*, *E. coli*, *K. pneumoniae*	No β-lactamase or DNA gyrase assays	[60]
Chlorpromazine, Thioridazine (Phenothiazines)	ROS induction, impaired DNA repair	In *E. coli*: ROS increase, mutagenesis, reduced DNA repair, higher antibiotic susceptibility	In vitro only	[61]
SSRIs (Fluoxetine, Sertraline, Paroxetine, Citalopram, Escitalopram)	ROS induction, DNA damage, immunomodulation	In MSSA, MRSA, *E. coli*, *K. pneumoniae*, *P. aeruginosa*: ROS increase, DNA strand breaks; checkerboard assays showed ROS potentiates β-lactam/fluoroquinolone efficacy	No β-lactamase effects; in vitro testing only	[62]
SSRIs (Fluoxetine, Sertraline)	ROS induction, immunomodulation	In macrophages infected with *S. aureus*: increased oxidative burst, improved killing; also, activity vs. *E. coli*, *K. pneumoniae*	No β-lactamase assays; no in vivo testing	[15]
SSRIs	ROS induction, DNA damage	In *S. aureus* and *E. coli*: ROS accumulation, DNA strand breaks, replication impairment	No β-lactamase assays	[17]
TCAs	ROS induction, DNA damage	In *S. aureus*, *E. coli*, *Klebsiella pneumoniae*, *Pseudomonas aeruginosa*: increased ROS, DNA fragmentation, growth suppression	No β-lactamase assays	[63]
Amlodipine (CCB)	Membrane perturbation, PBP interference	In *S. aureus*: membrane leakage, PBP interference; enhanced β-lactam activity	No specific enzyme assays	[64]
CCBs	Alter cell wall metabolism	Potentiated β-lactams; confirmed bactericidal activity	Mechanism unclear; no enzyme or ROS assays	[65]
Amlodipine (CCB)	Antibacterial activity	Inhibition zones vs. *S. aureus*, *E. coli*, *K. pneumoniae*, *P. aeruginosa*	No MIC or synergy data	[66]
Amlodipine (CCB)	Membrane perturbation, energy disruption	Dose- and time-dependent killing; synergy with thioridazine/promethazine	No mechanistic assays	[67]
Amlodipine (CCB)	β-lactamase inhibition	In MRSA: reduced β-lactamase activity, restored cefuroxime activity; FIC < 0.1 confirmed strong synergy	Limited to MRSA; not tested in Gram-negative β-lactamase producers	[68]
Statins	FMM disruption	In MRSA: disrupted membrane microdomains, destabilized resistance proteins	No mechanistic details; no MICs	[69]
Simvastatin (Statin)	Membrane perturbation, virulence pathway inhibition	Direct bactericidal activity vs. Gram-positives and Gram-negatives	Mechanism not fully defined	[70]
Statins	Antibacterial activity	Inhibited *Staphylococcus* and *Streptococcus* spp. from skin infections	In vitro testing only; no mechanism assays	[71]
Simvastatin (Statin)	Isoprenoid biosynthesis interference	Reduced *S. aureus* invasion of host cells via isoprenoid modulation	In vitro testing only	[72]
Simvastatin (Statin)	Host immunomodulation	In *S. aureus* skin wound infection: reduced burden, promoted healing	Animal model only; unclear direct vs. host effects	[73]
Simvastatin (Statin)	Mevalonate pathway disruption	In *S. aureus*: disrupted mevalonate metabolism, weakened cell wall integrity	In vitro testing only	[74]
Diclofenac (NSAID)	DNA replication interference	Suppressed DNA synthesis in Gram-positives/Gram-negatives; protected mice vs. *S. typhimurium*	Target undefined	[75]
NSAIDs (Carprofen, Bromfenac, Vedaprofen)	β-clamp inhibition (DNA Pol III)	In *E. coli*: inhibited β-clamp, disrupted replication	In vitro testing only	[76]
Diflunisal (NSAID)	β-clamp inhibition	In *H. pylori*: micromolar inhibitor, structurally confirmed binding at subsite I	In vitro only, limited testing scope	[77]
Naproxen (NSAID)	DNA intercalation + ROS (light-dependent)	On plasmid pBR322: ROS-mediated DNA nicking/cleavage under irradiation	Purified DNA only; physiological relevance unclears	[78]
Celecoxib (NSAID)	Host-directed immunomodulation (SIRT1 activation)	In *S. aureus*-infected macrophages: decrease in NF-κB and cytokines, increased oxidative enzymes; enhanced antibiotic killing	In vitro testing only	[79]
Celecoxib (NSAID)	Immunomodulation (microglia model)	In microglia infected with *S. aureus*: reduced bacteria, rebalanced cytokines, enhanced ciprofloxacin activity	In vitro testing only	[80]
Celecoxib (NSAID)	Direct synthesis inhibition (RNA, DNA, protein)	In *S. aureus*: inhibited RNA/DNA/protein synthesis; efficacy in *C. elegans* and murine MRSA skin model; synergistic with topical/systemic antibiotics	Gram-negative activity required colistin; topical testing only	[81]

Abbreviations: ROS, reactive oxygen species; MIC, minimum inhibitory concentration; FIC, fractional inhibitory concentration index; FMM, functional membrane microdomain; PBP, penicillin-binding protein; β-clamp, DNA polymerase III β subunit; NF-κB, nuclear factor kappa-light-chain-enhancer of activated B cells. Note: Most studies were conducted in vitro, often using single strains or laboratory models. No consistent evidence was found for direct β-lactamase inhibition, except for amlodipine in MRSA [65]. Findings highlight ROS-mediated DNA damage, β-clamp targeting, and immunomodulatory effects as key mechanisms, but clinical translation remains untested.

Another study demonstrated that heterocyclic antidepressants, including SSRIs, cause dose-dependent ROS accumulation and DNA damage in *S. aureus* and *E. coli* [17]. DNA integrity assays confirmed strand breaks and impaired bacterial replication [17]. In terms of tricyclic antidepressants (TCAs), evidence showed that TCAs produce similar antibacterial effects [63], with robust induction of ROS, DNA damage, and growth suppression in *S. aureus*, *E. coli*, and some Gram-negative clinical isolates of *K. pneumoniae* and *P. aeruginosa*. These experiments confirmed ROS production and DNA fragmentation but did not investigate β-lactamases.

Notably, many of these studies are yet to progress beyond the in vitro validation stage of research. These studies indicate potential for use to repurpose these compounds as new antibiotics, although they do not indicate whether the compounds would have similar effects in whole organisms. Bioavailability and biokinetic studies are required to assess the amount of the compound that is absorbed into the bloodstream, the levels of the drug that reach the target tissue, and how rapidly the drug is degraded. Additionally, studies are required to evaluate the toxicity of the end products of drug metabolism. In vivo testing using relevant animal models is required to more fully understand the druggable potential of these compounds as novel antibiotics. However, due to ethical considerations, such studies should only be performed after the compounds’ antibacterial activity and potency are validated in vitro.

Collectively, these studies show that antidepressants (both SSRIs and TCAs) exert antibacterial effects through ROS induction and oxidative stress, DNA damage and interference with replication, membrane perturbation, and immunomodulatory activation of host cells. Across all studies, no evidence was found for β-lactamase inhibition, as none used enzyme-specific tests such as nitrocefin hydrolysis or mass spectrometry of β-lactam degradation. Whilst these findings open intriguing avenues for drug repurposing, the current evidence remains early-stage, predominantly in vitro, and limited in strain diversity, highlighting the need for in vivo validation and clinical studies before translation.

#### 2.3.3. Calcium Channel Blockers

Evidence from recent studies highlight multiple antibacterial mechanisms for calcium channel blockers (CCBs). Amlodipine was shown to compromise the cell membrane integrity of *S. aureus*, causing leakage of intracellular contents [64]. It also interferes with penicillin-binding proteins, thereby enhancing β-lactam activity. Notably, no mechanism-specific assays were included, representing a key limitation in linking activity to enzymatic or cellular targets.

Similarly, calcium channel blockers and other non-antibiotic drugs have been shown to alter bacterial cell wall metabolism and potentiate β-lactam antibiotics, suggesting a mechanism distinct from classical antibiotic pathways [65]. That study confirmed bactericidal activity via microdilution assays, but mechanistic endpoints were not assessed, leaving enzyme inhibition, oxidative stress induction, and nutrient deprivation as unresolved possibilities [65]. Similarly, further evidence confirmed inhibition zones for amlodipine against *S. aureus*, *E. coli*, *K. pneumoniae*, and *P. aeruginosa*, yet they did not report MIC or synergy data, restricting conclusions about potency or clinical relevance [66].

Complementary findings were reported by another investigation where amlodipine displayed time and dose-dependent bactericidal effects [67], attributed to membrane perturbation and interference with energy-dependent processes, particularly when combined with other non-antibiotic agents such as thioridazine and promethazine. Yet, no mechanistic testing (enzyme inhibition, ROS, or cell division proteins) was undertaken, leaving functional interpretations incomplete.

The most mechanistically informative contribution comes from Yi et al. (2019), who performed both antimicrobial and enzymatic assays [68]. Amlodipine alone reduced MRSA growth at MICs of 32–64 µg/mL, but importantly, the drug inhibited β-lactamase activity in vitro, thereby restoring cephalosporin activity. When combined with cefuroxime, MICs dropped to 8 µg/mL and FIC values < 0.1 confirmed strong synergy attributable to β-lactamase inhibition [68]. This finding establishes the first direct mechanistic link between CCBs and enzymatic inhibition in major Gram-positive pathogens. However, the study did not extend to Gram-negative β-lactamase producers such as *K. pneumoniae* or *P. aeruginosa*, highlighting an important knowledge gap.

Taken together, these studies confirm that CCBs, particularly amlodipine, display in vitro antibacterial activity, depending on strain and experimental conditions. The enzymatic data from Yi et al. (2019) provide compelling evidence of β-lactamase inhibition, which may explain the observed synergy with β-lactams [68]. Other mechanisms, including DNA gyrase or topoisomerase inhibition, ROS induction, immunomodulatory potentiation, iron chelation, or disruption of bacterial cell division proteins, remain untested. Addressing these limitations through targeted mechanistic assays and in vivo models is essential to determine whether CCBs hold translational promise as antibiotic adjuvants.

#### 2.3.4. Statins

Statins, classically prescribed as HMG-CoA reductase inhibitors for hypercholesterolemia, have increasingly been shown to exert pleiotropic antibacterial effects. Statins have been shown to disrupt bacterial membrane microdomains in methicillin-resistant *Staphylococcus aureus* (MRSA), leading to disassembly of functional membrane microdomains (FMMs) that are critical for the stability of resistance-conferring proteins and thereby enhancing susceptibility to antibiotics [69]. This suggests that interference with lipid organization within bacterial membranes could be a key mechanistic pathway for overcoming drug resistance.

Further evidence showed that simvastatin exhibits direct bactericidal activity against both Gram-positive and Gram-negative pathogens [70]. That work suggested that the antibacterial activity may involve perturbation of membrane integrity and inhibition of virulence-associated pathways, independent of cholesterol-lowering effects in mammalian systems. Extending this line of evidence, several statins were reported to inhibit the growth of pathogens responsible for skin infections, including *Staphylococcus* and *Streptococcus* spp. [71]. These in vitro findings underscore the potential utility of statins as adjunct or repurposed agents against skin and soft tissue infections.

Mechanistic insights also point toward statin interference with isoprenoid biosynthesis. Another study revealed that simvastatin reduced the ability of *S. aureus* to invade host cells through modulation of isoprenoid intermediates [72]. By limiting the availability of these intermediates, statins may compromise bacterial processes that rely on prenylated proteins for virulence and intracellular persistence. Supporting these findings in vivo, simvastatin was shown to enhance wound healing in *S. aureus*-infected skin wounds. This effect was achieved not only by reducing the bacterial burden but also by modulating host immune and repair pathways [73].

Most recently, Cortês et al. (2025) provided direct biochemical evidence that simvastatin disrupts the mevalonate pathway in *S. aureus*, impairing cell wall integrity and rendering bacteria more vulnerable to environmental stresses [74]. This is particularly significant because the bacterial mevalonate pathway shares homology with the mammalian target of statins, suggesting a conserved mechanistic axis that could be exploited in antibacterial strategies.

Despite these advances, significant knowledge gaps remain. First, the extent to which different statins (lipophilic vs. hydrophilic) vary in their ability to disrupt bacterial membranes or mevalonate metabolism remains poorly defined [68,69,70,71,72,73,74]. Second, whilst functional membrane microdomain (FMM) disruption and cell wall weakening have been demonstrated, the downstream signalling cascades within bacteria leading to resistance collapse are not fully elucidated. Third, the dual contribution of host-directed immunomodulation versus direct bacterial targeting remains difficult to disentangle, as highlighted by several studies [72,73]. Future studies should focus on clarifying structure–activity relationships among different statins and mapping the bacterial prenylated proteins most affected by isoprenoid pathway interference. In addition, evaluating their synergistic mechanisms with β-lactams and glycopeptides, alongside in vivo pharmacokinetic-pharmacodynamic (PK–PD) correlation studies, is essential to determine whether clinically achievable statin concentrations or local delivery strategies are needed.

#### 2.3.5. Non-Steroidal Anti-Inflammatory Drugs

Multiple lines of evidence implicate bacterial DNA replication as a key target of NSAIDs. Diclofenac provided the earliest signal that NSAIDs can suppress bacterial DNA replication; in vitro, it inhibited DNA synthesis across Gram-positive and Gram-negative bacteria (via reduced [H]-thymidine incorporation), and it protected mice challenged with *Salmonella typhimurium* in vivo, although the precise replication target was not identified [75]. Building on this replication theme, several NSAIDs (carprofen, bromfenac, and vedaprofen) were shown to exert antibacterial activity by inhibiting the *E. coli* DNA polymerase III β subunit (β-clamp, DnaN), a central interaction hub for polymerase/loader partners [76]. Clamp-target engagement was confirmed by blocking interactions of the clamp loader and/or the replicative polymerase α subunit with the sliding clamp in an in vitro replication assay [76]. Extending β-clamp targeting to a different species, diflunisal was identified through in-silico screening and biochemical validation as a micromolar inhibitor of *Helicobacter pylori* growth [77]. It binds the *H. pylori* β-clamp at the subsite I protein–protein interaction pocket, with structural confirmation, thereby functionally disrupting replication processivity. In a distinct mechanism category, evidence established that naproxen intercalates DNA and, upon white-light irradiation, generates reactive oxygen species that nick plasmid DNA (pBR322) and cleave nucleic acids [78]. Methods included spectroscopy and agarose gel analysis on purified DNA rather than whole bacteria, indicating a chemically plausible, light-dependent ROS mechanism.

Host-directed activity is highlighted in two celecoxib studies. Annamanedi & Kalle (2014) used RAW264.7 macrophages infected with *S. aureus* ATCC 29213 and showed that celecoxib (10 µM) alone or with ampicillin (5 µg/mL) activated SIRT1, reduced TLR2–JNK–NF-κB signaling, lowered NO and pro-inflammatory cytokines (IL-6, MIP-1α, IL-1β), increased catalase/peroxidase, and thereby sensitized intracellular *S. aureus* to antibiotic killing [79]. In primary murine microglia cultures infected with live *S. aureus*, Dey et al. (2018) found that celecoxib combined with ciprofloxacin reduced viable bacteria and attenuated oxidative damage while re-balancing cytokines toward an IL-10-high, anti-inflammatory profile, thus linking immunomodulation with enhanced antibacterial effect under CNS-relevant conditions [80]. Finally, Thangamani et al. (2015) demonstrated direct antibacterial activity of celecoxib against *S. aureus* ATCC 29213, mapped its primary mechanism to dose-dependent inhibition of RNA, DNA, and protein synthesis via radiolabelled precursor incorporation assays (with secondary effects on lipid synthesis at higher doses), and established topical in vivo efficacy in MRSA infection models (*Caenorhabditis elegans* infected with MRSA USA300; murine MRSA skin infection) [81]. Importantly for combinations, celecoxib showed synergy against multidrug-resistant *S. aureus* strains (e.g., MRSA300, NRS119, NRS107, VRSA5) when combined with several topical agents (mupirocin, fusidic acid, bacitracin, retapamulin) and systemic antibiotics (gentamicin, clindamycin, rifampin, linezolid) [81]. For Gram-negative bacteria, activity was observed only when outer-membrane permeability was compromised using the permeabilizer colistin. Together, these results place replication-enzyme interference (notably β-clamp inhibition), ROS-linked DNA damage (photochemical with naproxen), and immune reprogramming (SIRT1-mediated) as the dominant NSAID mechanisms represented in this set, with combinational benefits documented for ampicillin, ciprofloxacin, and the above topical/systemic agents where specified [75,76,77,78,79,80,81].

Evidence for β-clamp targeting is strong but limited to selected NSAIDs and species such as *E. coli* and *H. pylori*. However, in vivo target engagement and PK–PD data remain scarce, and diclofenac’s replication blockade still lacks a defined protein target [75,76,77]. Naproxen’s ROS mechanism requires light and was demonstrated on purified DNA, leaving physiological relevance in infections uncertain [78]. Celecoxib has shown immune-modulating and direct antibacterial effects in macrophage/microglia models, *C. elegans*, and a topical mouse model [79,80,81]. Careful translation to human dosing and pathogen breadth is still required. Its Gram-negative activity appears to depend on permeability enhancement, such as with colistin. Across these studies, no experimental support appears for iron chelation/nutrient deprivation, β-lactamase inhibition, gyrase/topoisomerase binding, or direct inhibition of cell-division proteins: mechanisms that remain unaddressed here. Priority studies should therefore (i) broaden β-clamp target validation and resistance-liability mapping across pathogens, with in vivo exposure/engagement; (ii) test ROS-linked effects in whole-cell systems under infection-relevant conditions; (iii) quantify the contribution of SIRT1-centered immunomodulation to bacterial clearance and antibiotic potentiation; and (iv) systematically evaluate topical/local delivery (as with celecoxib) to overcome potency/exposure limits, whilst minimising systemic risk [76,77,78,79,80,81].

In addition, NSAIDs (diclofenac) has been shown to generate oxidative stress and trigger the bacterial SOS response, resulting in mutagenesis and rapid emergence of resistant phenotypes [82]. These oxidative and genotoxic effects can promote cross-resistance and may disturb commensal microbial communities. Thus, while NSAIDs like diclofenac may exhibit antibacterial or synergistic actions under specific conditions, their unintended capacity to induce ROS/SOS responses underscores the need for careful evaluation before considering them as antimicrobial adjuvants.

## 3. Artificial Intelligence in Antimicrobial Discovery: Current Advances and Translational Opportunities

Artificial intelligence (AI) has rapidly emerged as a transformative force in antimicrobial discovery, enabling the identification of novel compounds, elucidation of mechanisms of action (MOA), and the acceleration of drug development pipelines. The seminal study by Stokes et al. (2020) pioneered the integration of deep learning into antibiotic discovery, employing a message-passing neural network trained on growth inhibition data to identify *halicin*, a structurally distinct molecule with broad-spectrum activity, including efficacy against multidrug-resistant pathogens, and in murine infection models [83]. This work not only validated the ability of AI to generalise beyond traditional antibiotic scaffolds, but also introduced a scalable framework for mining vast chemical libraries. Building on this foundation, another study demonstrated that machine learning can guide pathogen-specific antibiotic discovery. That study identified *abaucin*, a narrow-spectrum antibiotic with activity restricted to *A. baumannii*, which functions through disruption of lipoprotein trafficking [84]. This represented a critical paradigm shift: AI can now direct the design of therapeutics that minimise microbiome disruption, whilst targeting high-priority pathogens. Complementing these advances, further evidence expanded discovery into the “microbial dark matter,” applying machine learning to global metagenomic datasets and predicting nearly one million antimicrobial peptides (AMPs). Experimental validation confirmed activity for dozens of candidates, several of which displayed in vivo efficacy [85]. This study underscores the power of AI to unlock previously inaccessible natural reservoirs of antimicrobial agents at an unprecedented scale.

Beyond discovery, AI has shown immense potential in mechanism of action deconvolution, a longstanding bottleneck in antimicrobial development. Espinoza et al. (2021) applied explainable machine learning to bacterial transcriptomic responses, achieving >99% accuracy in predicting antibiotic MOA across multiple compound classes and successfully flagging novel candidates with unique signatures [86]. Similarly, another investigation adapted bacterial cytological profiling to *M. tuberculosis*, leveraging convolutional neural networks to classify compound-induced morphological changes with high accuracy. This approach provides a rapid, label-free platform for mechanism prediction in one of the most challenging pathogens [87].

Crucially, the translation of these advances aligns with contemporary clinical challenges. The strains of concern in global health, *E. coli* (ESBL-producing), *K. pneumoniae* (ESBL-producing), *S. aureus* (including MRSA), *Shigella flexneri*, *Shigella sonnei*, *S. typhimurium*, and *Bacillus cereus*, are emblematic of the rise of multidrug resistance. Conventional antibiotics such as penicillin G, erythromycin, tetracycline, chloramphenicol, ciprofloxacin, polymyxin B, oxacillin, amoxicillin, vancomycin, and gentamicin are increasingly compromised by efflux pumps, enzymatic inactivation, and altered target sites. AI offers the possibility to systematically match novel or repurposed drugs to resistance phenotypes, identify adjuvant combinations, and prioritise agents that circumvent dominant resistance mechanisms.

Furthermore, AI frameworks provide a systematic means to match novel or rediscovered compounds, including non-antibiotic medicines, to pathogen-specific resistance profiles, thereby enabling the prediction of whether these agents act through efflux pump inhibition, membrane disruption, DNA/RNA synthesis interference, cell wall biosynthesis inhibition, or β-lactamase inactivation. The integration of transcriptomic and cytological profiling pipelines into these models allows for rapid mechanistic annotation, which is crucial for distinguishing true antimicrobial effects from secondary cellular perturbations. Such approaches not only accelerate the rational selection of compounds for synergy testing with existing antibiotics but also enable the identification of compounds with dual or multi-target actions that may suppress the emergence of resistance.

Collectively, these advances and translational opportunities highlight that AI is no longer an auxiliary tool, but rather a central engine in antimicrobial research. By uniting predictive discovery, pathogen-specific targeting, large-scale peptide mining, and automated mechanism of action classification with drug-repurposing hypotheses, AI offers a credible pathway to overcome decades of stagnation in antibiotic development. The convergence of these strategies provides not only a pipeline of novel chemical matter but also a rational framework for repositioning clinically approved medicines, facilitating precision-targeted therapy and synergy-based interventions against urgent bacterial threats. This evolving paradigm marks the beginning of an era in which AI-driven discovery and repurposing have the potential to reshape the global response to antimicrobial resistance.

However, significant challenges remain. Most non-antibiotic compounds display modest intrinsic antibacterial activity, and their therapeutic benefit is often context-dependent, requiring combination regimens. Furthermore, issues of toxicity, achievable concentrations in vivo, and off-target effects limit clinical translation. AI-driven predictions are ultimately constrained by the quality and diversity of training data, making experimental validation indispensable. Thus, whilst AI accelerates the identification of non-antibiotic scaffolds with antimicrobial potential, careful prioritisation and translational de-risking remain critical.

Furthermore, there is a paucity of data available for some diseases. As AI modelling requires large, high-quality datasets for accurate predictions, this is a substantial limitation for AI use at present. The development of better datasets is expected to address this issue in the future, providing greater benefits of using AI in repurposing non-antibiotic drugs as antibiotics. However, the volume and complexity of this data provide their own limitations due to current challenges in processing the data. Rapid advances in information handling through advanced computing power may address this issue in the future.

As AI use in drug discovery is still and emerging field, there is a relative lack of method standardisation between research groups, which many result in lack of consensus between research studies. Finally, due to its novel nature, there are multiple ethical considerations related to AI use in drug repurposing that are yet to be addressed, including data privacy, adherence to regulatory hurdles and algorithm bias towards specific populations within a dataset. However, this is a fast-moving and dynamic, and it is likely that future efforts will address these issues.

## 4. Conclusions

Non-antibiotic drugs, including antipsychotics, selective serotonin reuptake inhibitors (SSRIs), calcium channel blockers (CCBs), statins, and non-steroidal anti-inflammatory drugs (NSAIDs), are emerging as valuable candidates in the search for novel antimicrobial strategies. These agents exert antibacterial effects through diverse mechanisms including efflux pump inhibition, biofilm suppression, membrane disruption, DNA and enzyme inhibition, nutrient deprivation, and immunomodulatory activity. By targeting bacterial survival pathways distinct from traditional antibiotics, they hold promise, both as stand-alone agents, and as synergistic partners with existing therapies. Despite encouraging preclinical evidence, significant gaps remain regarding in vivo efficacy, optimal dosing, and long-term safety. Indeed, whilst most of these drugs were extensively evaluated for safety before their use was permitted clinically, they should be reevaluated in the context of their proposed use as antibiotics. Clinical drug toxicity is linked to many factors, including the method and site of administration, and the duration of treatment. Thus, the safety of a drug for its initially intended use does not necessarily show that it is safe to use as an antibiotic. All these compounds need substantially more toxicity testing. The integration of artificial intelligence into antimicrobial discovery offers powerful opportunities to systematically identify, optimize, and repurpose such agents, accelerating their translation into clinical practice. Harnessing the antimicrobial potential of non-antibiotics, combined with AI-driven discovery, could meaningfully expand our therapeutic arsenal against multidrug-resistant infections.

## Figures and Tables

**Figure 1 ijms-26-09880-f001:**
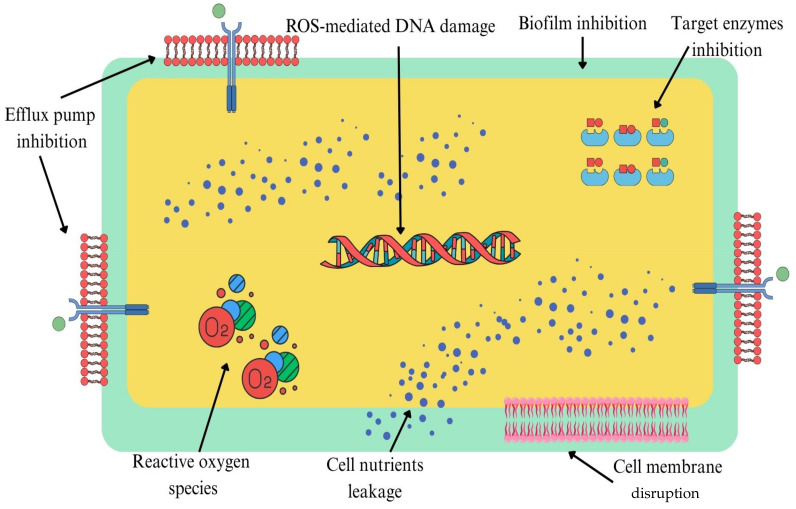
Possible antibacterial mechanisms of action of non-antibiotic drug classes. These mechanisms include inhibition of efflux pumps, suppression of biofilm formation, inhibition of target enzymes, leakage of essential cellular nutrients, disruption of the cell membrane, induction of reactive oxygen species (ROS) generation and ROS-mediated DNA damage.

**Table 1 ijms-26-09880-t001:** Non-antibiotic drugs with efflux pump inhibition activity and antibacterial synergy.

Drug (Class)	Possible Mechanism	Key Findings	Limitations	Ref
Thioridazine (AP, phenothiazine)	Efflux pump inhibition in *M. tuberculosis*	Reversed resistance to isoniazid and rifampicin; synergistic with first-line TB drugs	No MIC or FIC values provided	[7]
Thioridazine & Chlorpromazine (APs, phenothiazines)	Efflux pump inhibition in *M. tuberculosis*	Reduced clarithromycin and isoniazid MICs	No MIC/FIC data reported	[8]
Phenothiazines (APs)	Efflux pump inhibition in *S. aureus*	Enhanced oxacillin efficacy	No MIC data reported	[9]
Thioridazine (AP)	Efflux pump inhibition in *M. avium*	Reduced ethambutol MIC in (8 → 2 µg/mL); described as synergistic	No FIC index calculated	[10]
Chlorpromazine (AP)	Efflux inhibition in *S. aureus*	Reduced norfloxacin MIC (4 → 1 µg/mL); synergy inferred	No checkerboard synergy assays performed	[11]
Phenothiazines (APs)	Efflux pump inhibition in Gram-negatives and -positives	Synergy with penicillin against resistant bacteria; MIC as low as 2 µg/ml	No FIC indices calculated	[12]
Chlorpromazine (AP)	AcrAB-TolC efflux inhibition in *E. coli*	Reduced tetracycline MIC	No MIC/FIC values reported	[13]
Chlorpromazine (AP)	Efflux inhibition in *B. pseudomallei*	Synergy with erythromycin confirmed (FIC ≤ 0.5)	-	[14]
Sertraline (SSRI)	Efflux pump inhibition in *C. albicans*	Synergy with fluconazole; MIC90 = 3 µM; FIC < 0.5	Methods for FIC derivation not detailed	[15]
Fluoxetine (SSRI)	Possible efflux pump inhibition in Gram-negatives	MICs 15–126 µg/mL	No FIC values; synergy not assessed	[16]
Paroxetine (SSRI)	Efflux pump inhibition in *S. aureus*	MIC 64 µg/mL; enhanced aminoglycoside efficacy	No FIC values; synergy inferred only from MIC shift	[17]
Citalopram (SSRI)	Efflux pump inhibition (predicted)	Increased antibiotic susceptibility	No MIC/FIC data; based on pathway predictions, not direct assays	[18]
Escitalopram (SSRI)	Efflux pump inhibition in multidrug-resistant strains	Synergy with sulfamethoxazole–trimethoprim; MIC reductions observed	No FIC indices provided; synergy inferred from MIC shift	[19]
Verapamil (CCB)	Efflux pump inhibition (NorA) in *S. aureus*	Enhanced ciprofloxacin activity	No precise MIC or FIC values detailed	[20]
Verapamil (CCB)	Efflux pump inhibition in *M. tuberculosis*	Synergy with rifampicin	Detailed MIC reductions not provided	[21]
Verapamil (CCB)	Efflux pump inhibition in *M. tuberculosis* H37Rv	No significant synergy with isoniazid (FIC ≥ 0.5; additive/indifferent)	-	[22]
Verapamil (CCB)	Efflux inhibition ± membrane energetics disruption *M. tuberculosis*	Bedaquiline MIC 0.5 → 0.025 µM (20×); FIC 0.06; synergy with clofazimine (MIC 1.0 → 0.25 µM; FIC 0.19)	Mechanistic ambiguity; cardiotoxicity risk	[23]
Verapamil (CCB)	Likely membrane destabilisation/energy depletion rather than specific pump targeting in *M. abscessus*	Potentiated bedaquiline activity	No MIC/FIC numbers reported; mechanism not confirmed as direct efflux inhibition	[24]
Simvastatin (Statin)	Efflux pump inhibition + membrane disruption in *S. aureus*	MIC 15.6–31.25 µg/mL; synergistic with tetracycline (FIC < 0.5)	No in vivo validation	[25]
Simvastatin & Atorvastatin (Statins)	Efflux pump inhibition in Gram-positives	MIC 64–128 µg/mL; synergy with gentamicin/ciprofloxacin mostly additive (FIC > 0.5 ≤ 1)	Limited potentiation	[3]
Simvastatin (Statin)	Efflux inhibition in *P. aeruginosa*	MIC 32 µg/mL; synergy with levofloxacin (FIC 0.31)	Mechanistic details are limited	[26]
Atorvastatin (Statin)	Efflux pump inhibition (with Ru-complex) in *S. aureus*	MIC >128 → 4 µg/mL in combination; FIC 0.17–0.5	Mechanism not studied; ruthenium complex role unclear	[27]
Rosuvastatin (Statin)	Efflux/membrane modulation in *S. aureus*	With levofloxacin: MIC 4 → 0.5 µg/mL; FIC 0.3	No in vivo PK/Tox data	[28]
Rosuvastatin (Statin)	Efflux/membrane affects Gram-positives and Gram-negatives	With cefixime: 2–4-fold MIC reduction; FIC 0.37–0.49	Lack of in vivo validation	[29]
Statins + Azoles	Fungal efflux inhibition in *C. auris*	MIC reductions; FIC 0.5–1.0 (additive)	Mechanism hypothesised	[30]
Atorvastatin (Statin)	Efflux inhibition in *T. rubrum*	MIC 64 µg/mL; synergy with terbinafine (FIC 0.45)	Limited mechanistic validation	[31]
Ibuprofen & Diclofenac (NSAIDs)	Proposed efflux pump inhibition	MICs 64–512 µg/mL; synergy with gentamicin/ciprofloxacin (FIC < 0.5)	No efflux assays performed	[32]
Ibuprofen, Diclofenac, Aspirin (NSAIDs)	Proposed efflux pump inhibition in *S. aureus*	MICs 125–500 µg/mL; synergy with cefuroxime/chloramphenicol (FIC 0.313–0.625)	Mechanism inferred; no direct efflux assays	[33]
Celecoxib (NSAID)	Membrane disruption/efflux pump in *S. aureus*	MIC 32 µg/mL; synergy with oxacillin (FIC 0.25)	No efflux assays performed	[34]
Ibuprofen, Aspirin (NSAIDs)	Proposed efflux inhibition in *S. aureus*	Decreased oxytetracycline MIC 8 → 1 µg/mL; FIC 0.25	No validation of efflux inhibition	[35]
Diclofenac (NSAID)	Proposed efflux inhibition in Gram-positives and Gram-negatives	MIC 64 µg/mL; synergy with gentamicin (FIC 0.38)	No direct efflux assays	[36]
Diclofenac + Essential oils (NSAID + EO)	Membrane disruption, ROS in *Candida* spp.	Synergy (FIC < 0.5)	No efflux assays	[37]
Ibuprofen (NSAID)	Biofilm formation and efflux pump gene expression in *P. aeruginosa*	MIC 2048 µg/mL; synergy with ciprofloxacin (FIC 0.4)	Suppressed efflux pump gene expression and virulence factors	[38]
Ketorolac (NSAID)	Biofilm matrix modulation *S. aureus* biofilms	Synergy with gentamicin (FIC 0.31)	No efflux assays performed	[39]

Abbreviations: MIC, minimum inhibitory concentration; FIC, fractional inhibitory concentration; TB, tuberculosis; PK, pharmacokinetics; Tox, toxicity; EO, essential oils; ROS, reactive oxygen species; AP = antipsychotic; CCB, calcium channel blocker; SSRI, selective serotonin reuptake inhibitor; NSAID, non-steroidal anti-inflammatory drug; Ref = references cited. “Synergy” was defined as an FIC index ≤ 0.5, while additive effects correspond to FIC values > 0.5 to ≤1.0. Many studies reported qualitative MIC reductions without detailed synergy validation, which limits reproducibility and clinical translation. - indicates not applicable.

**Table 2 ijms-26-09880-t002:** Non-antibiotic drugs with biofilm inhibition or membrane-disruptive activity.

Drug (Class)	Possible Mechanism	Key Findings	Limitations	Reference
Thioridazine (Antipsychotic, phenothiazine)	Biofilm disruption via efflux inhibition	Disrupted biofilms of *M. tuberculosis* and *M. ulcerans*; potentiated antibiotics	No MBIC/MBEC, MIC/FIC data; no membrane assays (PI, SEM, TEM)	[40]
Chlorpromazine (Antipsychotic, phenothiazine)	Biofilm reduction via efflux inhibition	Reported activity in *S. aureus* biofilms (NorA-related)	No MBIC/MBEC; no membrane assays	[41]
Chlorpromazine (Antipsychotic, phenothiazine)	Biofilm reduction via efflux inhibition	Reduced biofilm resilience in *S. aureus*	No MBIC/MBEC; no membrane assays	[42]
Branched polyethylenimine (BPEI, polymer)	PBP2a inhibition, biofilm disruption	Restored *S. epidermidis* (MRSE) susceptibility to amoxicillin; also affected MRSA, *P. aeruginosa*, *E. coli*	No MIC/MBIC reductions reported; limited synergy quantification	[43]
SSRIs (Fluoxetine, Sertraline, Paroxetine, Fluvoxamine)	Biofilm/membrane disruption	Increased membrane permeability vs. ESKAPEE pathogens	No MBIC/MBEC or % biofilm data; no membrane assays beyond dye uptake	[16]
Fluoxetine (SSRI)	Proposed biofilm inhibition via membrane disruption	With ampicillin, vancomycin, linezolid: synergy vs. *S. aureus* (incl. MRSA) and *E. faecalis*	No MIC/MBIC, FIC data	[44]
SSRIs (Fluoxetine, Sertraline)	Antimicrobial activity	Reported effects in *E. coli* and fungi (*Candida albicans*, *C. glabrata*, *C. parapsilosis*, *C. tropicalis*)	No MBIC/MBEC, no quantitative biofilm data, no membrane assays	[45]
SSRIs	Biofilm disruption	Reported vs. *Pseudomonas*, *Aeromonas*, *Enterobacter* spp.	No quantitative biofilm or mechanistic data	[46]
Simvastatin (Statin)	Antibiofilm effect, membrane interference	Inhibited biofilm formation and reduced viability of *S. aureus* (crystal violet, CFU)	No MBIC/MBEC; no membrane assays; no biofilm synergy data	[47]
Statins	Antibiofilm effect	Tested vs. *S. aureus* UAMS-1 biofilms (MBEC assay)	No MBEC numbers; no synergy testing; no membrane assays	[48]
Ibuprofen & Diclofenac (NSAIDs)	Biofilm inhibition	Reduced biofilm in *E. coli* and *K. pneumoniae* UTI isolates; diclofenac ~50% reduction at 50 mg/L; ibuprofen suppressed morphotypes	No MBIC/MBEC defined; mechanistic evidence indirect	[49]
Ibuprofen & Diclofenac (NSAIDs)	Biofilm inhibition	Reduced *S. aureus* and *E. coli* biofilms on mesh; SEM confirmed fewer adherent cells	No MBIC/MBEC defined	[50]
Piroxicam, Diclofenac, Acetylsalicylic acid (NSAIDs)	Biofilm inhibition	Piroxicam reduced metabolic activity and removed biofilm mass in *E. coli* and *S. aureus* preformed biofilms	No MBIC/MBEC; MBC >2000 µg/mL; limited mechanistic data	[51]
NSAIDs (general)	Biofilm disruption via QS and virulence interference	Downregulated AgrA in *S. aureus*; interfered with quorum sensing, altered surface properties	No MBIC/MBEC; no membrane assays	[52]
Salicylic acid (NSAID metabolite)	Iron chelation, biofilm modulation	Enhanced PIA-mediated biofilm synthesis in *S. aureus* via Fe^2+^ limitation	Adverse effect; no MBIC/MBEC values	[53]

Abbreviations: MBIC, minimum biofilm inhibitory concentration; MBEC, minimum biofilm eradication concentration; MIC, minimum inhibitory concentration; FIC, fractional inhibitory concentration; MBC, minimum bactericidal concentration; PI, propidium iodide uptake assay; SEM, scanning electron microscopy; TEM, transmission electron microscopy; CFUs, colony-forming units; QS, quorum sensing; MRSA, methicillin-resistant *Staphylococcus aureus*; MRSE, methicillin-resistant *Staphylococcus epidermidis*; UTI, urinary tract infection; PIA, polysaccharide intercellular adhesin. Notes: No study consistently reported MBIC or MBEC values, and most evidence is qualitative or indirect. Where synergy is mentioned, no formal FIC indices or quantitative reductions were provided.

## Data Availability

No new data were created or analyzed in this study. Data sharing is not applicable to this article.

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
