# Peer review of "Beyond Antibiotics: Repurposing Non-Antibiotic Drugs as Novel Antibacterial Agents to Combat Resistance"

_ijms, 2025, doi:10.3390/ijms26209880_

Round 1
Reviewer 1 Report
Comments and Suggestions for Authors
This review comprehensively summarizes the potential of repurposing non-antibiotic drugs (e.g., NSAIDs, statins, SSRIs) to combat antimicrobial resistance (AMR), covering their antibacterial mechanisms, synergistic effects with antibiotics, and the role of AI in discovery. The topic is highly relevant to current clinical needs, and the structure is clear, with detailed subsections on key drug classes and mechanisms.
In Section 3, while the application of AI in antimicrobial discovery is introduced, there is no discussion on the limitations of AI models in repurposing non-antibiotic drugs. Could the authors add relevant content to objectively evaluate the current challenges of AI in this field and avoid overestimating its practical application value?
In Table 3, several entries lack in vivo validation data. The review mentions this limitation generally, but could the authors specifically comment on which of these in vitro mechanisms are most urgently needed to be validated in animal models, based on their potential to address high-priority AMR pathogens (e.g., MRSA, M. tuberculosis)?
Moreover, in Table 1-3, are there any studies in the literature regarding biological safety or side effects? If applicable, it should also be listed in the "limitations" section.
Reviewer 2 Report
Comments and Suggestions for Authors
See attached.
